

# Joint assimilation of soil moisture retrieved from multiple passive microwave frequencies increases robustness and quality of soil moisture state estimation

Anouk I. Gevaert[1], Luigi J. Renzullo[2,3], Albert I. J. M. van Dijk[3], Hans J. van der Woerd[4], Albrecht H. Weerts[5,6], Richard A. M. de Jeu[7]

[1] Earth and Climate Cluster, Department of Earth Sciences, VU University Amsterdam, Amsterdam, the Netherlands
[2] CSIRO Land and Water, Canberra, Australia
[3] Fenner School of Environment and Society, Australia National University, Canberra, Australia
[4] Institute for Environmental Studies (IVM), VU University Amsterdam, the Netherlands
[5] Deltares, Delft, the Netherlands
[6] Department of Environmental Sciences, Wageningen University, the Netherlands
[7] VanderSat B.V., Haarlem, the Netherlands

*Correspondence to*: Anouk I. Gevaert (a.i.gevaert@vu.nl)

**Abstract.** Soil moisture affects the partitioning of water and energy and is recognized as an essential climate variable. Soil moisture estimates derived from passive microwave remote sensing can improve model estimates through data assimilation, but the relative effectiveness of microwave retrievals in different frequencies is unclear. Land Parameter Retrieval Model (LPRM) satellite soil moisture derived from L-, C- and X-band frequency remote sensing were assimilated in the Australian Water Resources Assessment landscape hydrology model (AWRA-L). Two sets of experiments were performed. First, each retrieval was assimilated individually for comparison. Second, each possible combination of two retrievals was assimilated jointly. Results were evaluated against field-measured top-layer and root-zone soil moisture at 24 sites across Australia. Assimilation generally improved the coefficient of correlation ($r$) between modelled and field-measured soil moisture. L- and X-band retrievals were more informative than C-band retrievals, improving $r$ by an average of 0.11 and 0.08 compared to 0.04, respectively. Although L-band retrievals were more informative for top-layer soil moisture in most cases, there were exceptions, and L- and X-band were equally informative for root-zone soil moisture. The consistency between L- and X-band retrievals suggests that they can substitute for each other, for example when transitioning between sensors and missions. Furthermore, joint assimilation of retrievals resulted in a model performance that was similar to or better than assimilating either retrieval individually. Comparison of model estimates obtained with global precipitation data and with higher-quality, higher-resolution regional data, respectively, demonstrated that precipitation data quality does determine the overall benefit that can be expected from assimilation. Further work is needed to assess the potentially complementary spatial information that can be derived from retrievals from different frequencies.



## 1 Introduction

Soil moisture plays an important role in the water and energy cycles, as it controls the partitioning of rainfall into evaporation, infiltration, and runoff (Seneviratne et al., 2010). For this reason, soil moisture observations have great potential to improve the performance of land surface models. There are various ways that soil moisture observations can be used to

improve models, including in model calibration (e.g. Wooldridge et al. 2003; Wanders et al. 2014) and by constraining initial conditions (e.g. Jacobs et al. 2003; Massari et al. 2014). One of the most popular techniques to merge observational and model data is through data assimilation, which has been shown to improve model state estimates of surface and root-zone soil moisture (Renzullo et al. 2014; Draper et al. 2012; Reichle and Koster 2005), evaporation (Tian et al., 2017), and runoff (Brocca et al., 2010; López López et al., 2016).

A variety of observation-based soil moisture estimates have been used in data assimilation, including field- (Aubert et al. 2003; Lee et al. 2011), airborne- (Margulis et al., 2002) and space-borne (e.g. Reichle et al. 2007; Pauwels et al. 2001) measurements. Soil moisture datasets based on microwave remote sensing are of particular interest because of their global coverage, near-daily resolution and low sensitivity to atmospheric interferences. Several studies have assimilated satellite soil moisture retrievals derived from active (e.g. Pauwels et al. 2001; Brocca et al. 2011; Draper et al. 2011) and passive (e.g.

Reichle et al. 2007; Liu et al. 2011; Gao et al. 2007; López López et al. 2016)  microwave sensors, as well as a combination of passive and active sensors (Draper et al. 2012; Renzullo et al. 2014).

Aside from the distinction between active and passive microwave sensors, soil moisture retrievals can be retrieved from different frequencies. Common frequencies are L band (1.4 GHz), C band (6 GHz) and X band (10 GHz). Of these, L band is often assumed optimal for soil moisture retrieval because it is less sensitive to vegetation cover and the atmosphere than the

higher frequency retrievals, as well as having a deeper signal depth (e.g. Schmugge 1978; Ulaby et al. 1986). At the same time, the lower emission intensity means that observation footprint increases with decreasing frequency, creating a trade-off between spatial detail on the one hand, and observation depth and interference from vegetation and the atmosphere on the other. Studies evaluating and comparing satellite soil moisture retrievals have empirically confirmed that errors in soil moisture retrievals tend to increase with increasing frequency (Dorigo et al., 2010). Also, L-band retrievals tend to

outperform C-band retrievals over the more moderate vegetated regions (e.g. Holgate et al. 2016; Al-Yaari et al. 2014; van der Schalie et al. 2016). It may be expected that this higher accuracy leads to higher benefits of data assimilation, but this is yet to be demonstrated.

Ultimately, the benefit of assimilation depends on the relative magnitude of errors in the retrievals and model, respectively. Holgate et al. (2016) and Renzullo et al (2016) found that model estimates of soil moisture already had better accuracy than

those from remote sensing for some Australian sites, in which case there may be little benefit from assimilation. However, this may have been a function of the good quality precipitation estimates due to a relatively dense station measurement network. Precipitation estimates can be expected to be of considerably lower quality for many parts of the world, and there is a need to understand whether satellite data assimilation may be more beneficial under those circumstances.





Here, we assimilate passive microwave retrievals derived from three different frequencies, but using a common radiative transfer model. Our main objective is to understand differences between the retrievals in terms of their performance in data assimilation experiments and to investigate whether there is added value in their joint assimilation. In addition, we evaluate to what extent the benefit of assimilation depends on the quality of precipitation estimates used in modeling.

## 2 Soil moisture data

### 2.1 Satellite data

Soil moisture data were derived from brightness temperatures from two space-borne sensors. The Advanced Microwave Scanning Radiometer 2 (AMSR-2) provides data for the C- (6.9 GHz) and X-band (10.65 GHz) frequencies. These data have spatial resolutions of approximately 50 (C band) and 38 km (X band), respectively, and are sensitive to the top 1–2 cm soil layer (Owe et al. 2008). The Soil Moisture Ocean Salinity (SMOS) provides L-band (1.4 GHz) brightness temperatures. These observations have a spatial resolution of 43 km and are expected to be sensitive to the uppermost 5 cm of the soil. We focus on retrievals based on the nighttime overpasses (i.e. descending for AMSR-2, ascending for SMOS) because at night the assumption of equal vegetation and surface temperature is better met (de Jeu 2003; Liu et al. 2011). It has also been shown empirically to produce better results (Holgate et al., 2016). Soil moisture data were derived from the C-, L-, and X-band brightness temperatures using the Land Parameter Retrieval Model (LPRM, Owe et al. 2008) v6 with the parameterizations described in Parinussa et al. (2015) and van der Schalie et al. (2016). Despite the common retrieval models, the parameterizations vary with the frequency of the brightness temperatures. Lastly, all three datasets were resampled to a regular 0.25° grid.

### 2.2 Field-measured data

Field-measured top-layer and root-zone soil moisture data obtained from two networks, OzNet and OzFlux, are used as a benchmark. The OzNet network consists of 63 sites in southeastern Australia that measure soil moisture in the upper 5 or 8 cm of the soil up to 90 cm depth every 20 to 30 minutes (Smith et al., 2012). OzFlux (www.ozflux.org.au) consists of 36 sites in Australia and New Zealand where carbon, energy and water fluxes are measured. Three main criteria were applied for selecting suitable OzNet and OzFlux sites. First, only sites with at least 100 observations of top-layer soil moisture during the assimilation period were included. Second, only sites meeting the minimum number of triples for the triple collocation analysis were used. Finally, the data must be publicly available. Out of the 99 sites, 24 sites were selected. The sites consist of 12 OzNet sites and 12 OzFlux sites and cover a range of climate types based on a modified Köppen classification system developed by the Bureau of Meteorology (Fig. 1, details of sites in Table S1). The sensor measurements were converted to root-zone values based on weighted averages of the observed values, where weights correspond to the portion of the 90 cm root-zone layer closest to each sensor. Six OzFlux sites were excluded from the root-zone analysis because they had no sensors below 50 cm depth.





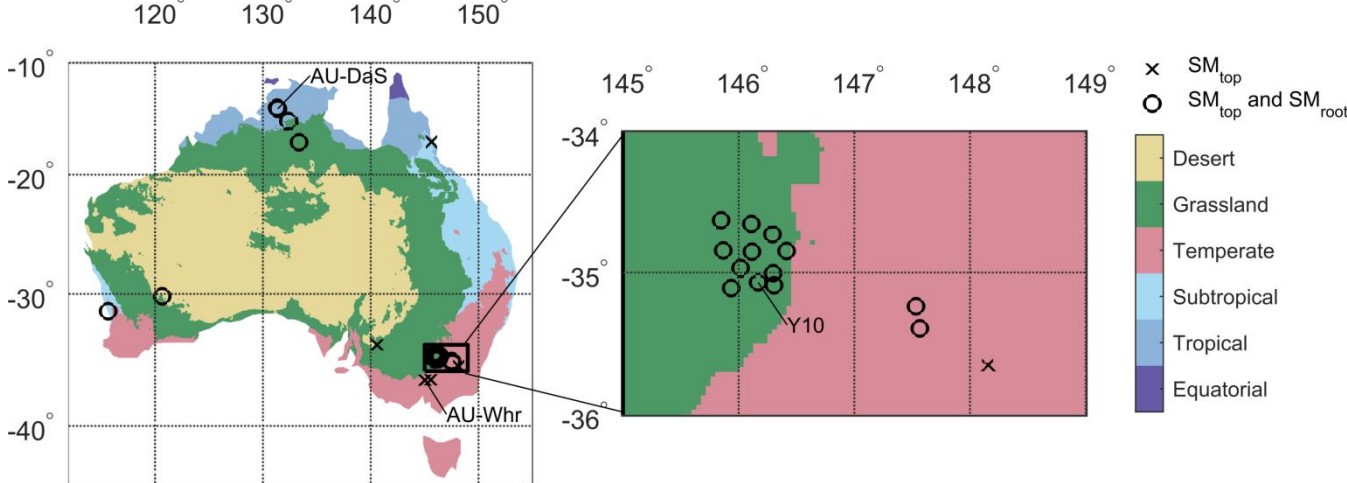

**Figure 1: Field-measured soil moisture stations from the OzNet and OzFlux networks plotted against the major Köppen Geiger climate zones. Stations indicated by crosses provide top-layer soil moisture data individually, sites indicated by a circle also provide root-zone data.**

## 3 Methods

### 3.1 Hydrological model

The landscape hydrology component of the Australian Water Resources Assessment system (AWRA-L) simulates the dynamics of hydrological states and fluxes at a continental scale (van Dijk, 2010) and is the model underpinning Australia's water resources assessments and accounts (Hafeez et al., 2015). The grid-based model has a 0.05° resolution and is run at a daily time step. It is important to note that each grid cell is modelled independently, meaning that there is no lateral exchange of water between neighboring cells.

AWRA-L consists of three soil layers that, in contrast to most land surface models, do not have a predefined depth (Fig. 2). Instead, each soil layer has a prescribed maximum water storage capacity. Soil wetness outputs are water storages relative to the available soil water, or the difference between wilting point and field capacity, and therefore range from 0 to 1. The water storage can be converted to volumetric water content when combined with soil texture data (Renzullo et al., 2014). Precipitation, reduced by interception and direct runoff, enters the soil column by the first soil layer. This top layer generally corresponds to a thickness of 5–10 cm and is also where soil evaporation occurs. The second layer is the shallow root layer and has a thickness of 10–20 cm. Conceptually, this layer is where shallow-rooted vegetation withdraws water for transpiration. The third soil layer, or deep root layer, has a thickness of 6–8 m and is accessed by deep-rooted vegetation only. Finally, there is an underlying groundwater store which can transfer water to the deep root layer by means of capillary rise. Runoff from the grid cell consists of the direct runoff from the surface and drainage from the groundwater store. The runoff can be used as an input for a routing model to calculate river discharge, but this is part of another AWRA model system component (Hafeez et al., 2015).



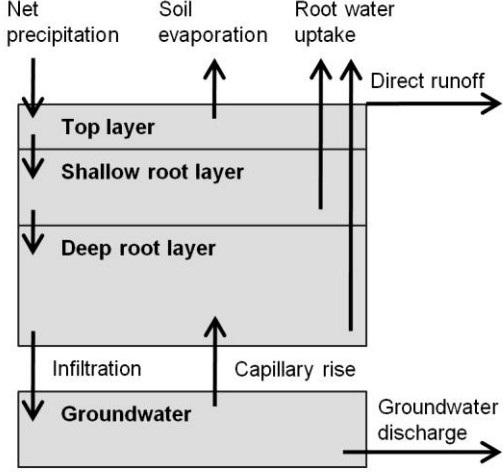

**Figure 2: Schematization of the structure of AWRA-L showing the three soil layers, groundwater store, and the relevant hydrological fluxes. Net precipitation refers to the precipitation reduced by interception.**

AWRA-L was run at a point scale, at locations where field-measured soil moisture data were available. The model was forced with radiation, daily minimum and maximum temperature from Australian Gridded Climate Data (AGCD) provided by the Bureau of Meteorology (Jones et al., 2009). These data have a 0.05° resolution. Two datasets with different spatial resolutions and quality were used to prescribe precipitation on a daily basis. Tropical Rainfall Measuring Mission (TRMM) 3B42 Research Version (GES DISC 2016) daily precipitation data with a 0.25° resolution were used for the main experiments. In an additional analysis, we use the model forced with higher-resolution gridded AGCD precipitation dataset as a benchmark. In this way, we assess whether precipitation data quality affects the potential for data assimilation to improve model performance. The assimilation experiments cover the period from July 2012 until April 2015, with model simulations starting in January 2005 to avoid errors in the initial conditions. The model version and parameterization (v0.5; van Dijk 2010) have not been calibrated with either satellite or field-measured soil moisture data.

### 3.2 Data pre-processing

Satellite soil moisture data were prepared for assimilation in several steps. First, satellite data were assigned to the model pixels using a nearest-neighbour approach. Next, the data were scaled to the model range to reduce bias. Previous studies have used a variety of methods for bias correction, such as linear rescaling between the minimum and maximum values (Brocca et al., 2011) or between model wilting point and field capacity (López López et al., 2016), first and second moment matching (Draper et al. 2009; Brocca et al. 2010), and cumulative density function matching (Reichle and Koster 2004; Draper et al. 2012). Each approach has its strengths and weaknesses. In this study, we apply a linear rescaling method matching the 5[th] and 95[th] percentiles of the observed dataset to the model data. We chose this approach because it retains the temporal distribution and scales it to the model dynamic range in a manner that avoids undue influence from outliers.





The errors for the three datasets were estimated using triple-collocation analysis (Dorigo et al., 2010; Scipal et al., 2008; Stoffelen, 1998). This analysis uses three datasets to quantify the error in each and operates under the assumption that the error structures are independent. One of the three datasets is chosen as a reference, and the other two datasets are rescaled to this dataset using linear rescaling factors provided by the triple collocation analysis. The errors are then produced in the data

space of the reference dataset. Advanced Scatterometer (ASCAT) soil moisture was used to complete the required triplet with the AWRA-L open loop (i.e. no assimilation) and each of the passive microwave estimates in turn. ASCAT soil moisture data are derived from C-band (5.3 GHz) radar observations using a change detection algorithm (Wagner et al. 1999; Naeimi et al. 2009). The data have a near-daily temporal resolution and a spatial resolution of 0.25°. Before applying the triple collocation analysis, the ASCAT data were rescaled using the same 5th–95th percentile scaling method that was applied

to the passive microwave data. The choice of reference dataset is arbitrary in triple collocation analysis, but by using AWRA-L as the reference dataset all errors were expressed in model wetness units.

In the joint assimilation experiments, we need to account for the difference in observation frequency of the SMOS and AMSR-2 soil moisture retrievals. On average, SMOS retrievals are available on 40 % of the days in the assimilation period, compared to nearly 75 % for AMSR-2 retrievals. Instead of subjectively adjusting the error estimates or weighting factors,

we aggregated the time series with a higher observation frequency to the measurement intervals of the time series with fewer observations. In other words, for each day with a measurement in the SMOS time series, we averaged all observations in the AMSR-2 time series that occurred between the previous SMOS observation and the current observation. We limited the aggregation period to a maximum of seven days and assigned equal weights to all observations falling within the aggregation period. Alternatively, more recent observations could be given more weight, but as there are usually only one or two AMSR-

2 observations between SMOS observations, we consider a simple average to be suitable. The errors assigned to the resulting concurrent observation time series were recalculated using the triple collocation method to use as weights in the joint assimilation experiments.

## 3.3 Assimilation procedure

Soil moisture data were assimilated using an Ensemble Kalman Filter (EnKF) approach. This is a relatively simple and

common method for assimilating observations into a variety of models (e.g. Draper et al. 2012; Reichle et al. 2002; Renzullo et al. 2014; López López et al. 2016) and consists of two steps. In the forecast step, ensembles are generated by perturbing the meteorological forcing data and propagating the model to the next time step. The ensembles are used to characterize the model error variances. In the analysis time step, the model states are adjusted towards the observations.

The analysis time step is calculated as

$$x_t^a = x_t^f + K_t[y_t - H_t(x_t^f)]$$   (1)



where $x_t^f$ is the model forecast, $x_t^a$ the model analysis, $K_t$ the Kalman gain, $y_t$ the observation and $H_t$ the observation model, which relates the model state to the observations, all at time $t$. The Kalman gain expresses the relative weighting of the observations with respect to the model and is defined as

$$K_t = P_t H_t^T (R_t + H_t P_t H_t^T)^{-1}, \qquad (2)$$

where $R_t$ is the observation error variance for a certain location, $H_t P_t H_t^T$ is the model error variance matrix, and $P_t H_t^T$ is the covariance matrix between the model states and model observations. Observational error is site-specific, but fixed in time. Model error variance is based on ensemble spread and varies over space and time (see below). If the model error is much lower than the observation error, $K_t$ will approach zero and the observation will not impact the model analysis. Alternatively, if the observation error is much lower than the model error variance, the model analysis will be dominated by the

observation. This assimilation updating procedure was applied to the first two soil layers of the AWRA-L model: the top layer and the shallow root layer.

In ensemble-based assimilation techniques, the ensemble spread must accurately represent the model error (Reichle et al. 2008; Turner et al. 2008). Especially after long periods with no rainfall, ensemble collapse can occur, which essentially prevents the observations from having any impact. One way to counter this is by applying a covariance inflation factor to the

model ensembles (Anderson and Anderson, 1999). Here, we applied a variable inflation factor to ensure a minimum model error of 2 %.

A total of 100 ensembles were generated by perturbing the rainfall, radiation and temperature data in following Renzullo et al. (2014). Precipitation errors were multiplicative and drawn from a uniform distribution ranging ±60 % of (i.e. 0.4–1.6 times) the forcing value. This error was based on spatial error estimates for the AGCD precipitation dataset (Jones et al.,

2009), but was also applied to the TRMM precipitation data. Radiation and temperature perturbations were additive, with assumed standard deviations of 2 K and 50 W m$^{-2}$, respectively. A correlation structure was enforced to minimize unlikely combinations of the radiation, temperature and rainfall perturbations (in that order), specified by Renzullo et al. (2014):

$$C = \begin{pmatrix} 1 & 0.7 & -0.8 \\ 0.7 & 1 & -0.5 \\ -0.8 & -0.5 & 1 \end{pmatrix}. \qquad (3)$$

### 3.4 Model evaluation

Model performance was based on the agreement between model soil moisture and field-measured data. Specifically, we based model performance on Pearson's $r$ between the model ensemble mean and daily averages of the field-measured time series during the assimilation period. The impact of data assimilation on model performance is defined as the difference between $r$ for the open loop and data assimilation scenarios, $\Delta r$. Other methods of evaluation, such as root mean square error and bias, were not included because AWRA-L simulates water storage in the soil layers rather than volumetric water content.

The correlations were calculated using the actual time series as well as anomaly values, i.e. the deviations from the





climatology. The climatology was calculated as the average of all days in the assimilation period (June 2012 to April 2015) falling within a 31-day window centered on a given day of the year. Correlations based on actual values reflect the ability of the modelled time series to capture the seasonal pattern of soil moisture, while $r$ for anomalies reflect the ability to capture deviations from the seasonal pattern. Significance levels of $\Delta r$ are based on the Steiger test for dependent correlations

(Steiger, 1980), using a significance level of $p < 0.05$.

The strong differences in the spatial representativeness is a complicating factor in the evaluation; for field measurements it is in the order of centimeters, the model has a grid resolution of approximately 5 kilometers, and the satellite data have a footprint of tens of kilometers diameter. Nevertheless, higher agreement with field-measured soil moisture increases confidence in model and satellite soil moisture estimates.

**3.5 Experimental setup**

Our main goal is to assess and compare the ability for soil moisture retrievals based on multiple passive microwave frequencies to improve the performance of the AWRA-L model. For reference, we compare model and satellite-based soil moisture to field-measured data as an indicator of the relative performance of the retrievals and of the potential for data assimilation to improve model estimates. Then, we address the main goal through three sets of experiments.

First, each of the three datasets was assimilated individually. The impact of data assimilation ($\Delta r$) was used to compare the results of the experiments, thus comparing model performance against field-measured data for the data assimilation scenarios using the open loop model performance as a reference. The model simulations were evaluated over the entire study period as well as split into the wet and dry seasons, where the wet season was defined as the six-month period with the highest average top-layer soil moisture content based on the open loop model simulation for each site. The results of this experiment are used

to evaluate and compare the ability of each of the retrievals to improve model performance.

Second, we assimilated each set of two soil moisture datasets (i.e. L-band and C-band, L-band and X-band, C-band and X-band retrievals) jointly. In this experiment, the impact of assimilation was again assessed based on $r$. However, the performance of the single retrieval assimilation experiments was used as a reference rather than the open loop simulation. In this way, we evaluate the added value of joint assimilation with respect to assimilating the retrievals individually.

Third, we repeated the single and joint assimilation experiments forcing the model with higher-resolution and higher-quality AGCD precipitation instead of TRMM precipitation. This experiment aimed to assess whether the quality of the precipitation data affects the impact of data assimilation on model performance. For reference, we compared open loop model performance using the two precipitation datasets as an indicator of the difference in precipitation quality. Then, we compared the impact of data assimilation for the two datasets, based on the change in model performance after data

assimilation ($\Delta r$).

All three experiments were designed from an empirical rather than theoretical point of view. We compare the effect of assimilating soil moisture retrievals based on multiple microwave frequencies, but additional factors would need to be taken into account to truly isolate the effect of frequency. First, despite the common retrieval model, parameterization of the model





such as surface roughness and single scattering albedo differ. In addition, AMSR-2 retrievals use simultaneously retrieved passive microwave observations to derive the soil temperature, while SMOS retrievals use model temperature (van der Schalie et al., 2016). Second, we did not correct for differences in the characteristics of the sensors and platforms, such as viewing angle, overpass time, and observation depth. Third, the error characteristics of the soil moisture retrievals used to

5    determine the weights of the observations in the assimilation scheme were allowed to vary between retrievals. This choice was deliberate, as applying a single error value to all retrievals may create a mismatch between the assigned error value and the quality of the observations. As a result, the difference in assimilation results reflects many characteristics of the retrievals and not only the frequency itself.

## 4 Results

10   We first assessed the potential for satellite soil moisture assimilation to improve the model based on a comparison against field observations. The model skill is variable, with $r$ between field-measured and top-layer soil moisture as high as 0.8 and as low as 0.3. The skill of the model is generally higher for actual values (Fig. 3a) than for anomalies (Fig. 3b). At most sites, L-band retrievals have the highest agreement with field-measured data, followed by the X- and C-band retrievals, respectively (Fig 3a). For anomaly time series, however, L- and X-band retrievals perform similarly well (Fig. 3b). The

15   results of the triple collocation analysis independently confirm these patterns, with lower errors for L- and X-band soil moisture and higher errors for C-band soil moisture (Fig. 3c).



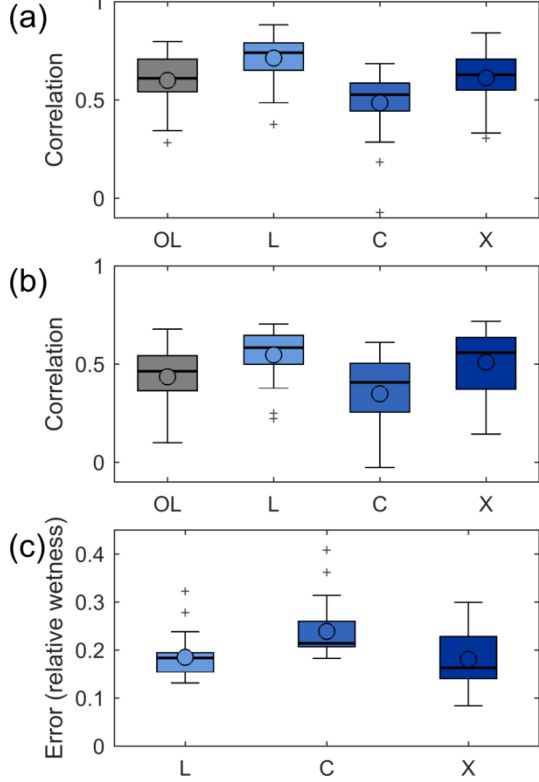

**Figure 3: Correlations (*r*) between the top-layer open-loop model soil moisture (OL) and satellite soil moisture observations against field measurements between July 2012 and April 2015 (a) and the same using anomaly time series (b). The triple collocation errors for the satellite soil moisture retrievals in AWRA-L wetness units are shown in (c). Box plots show the inter-quartile range (box), outliers (+), the median (thick line) and the mean (o). Outliers are based on points at a distance larger than 1.5 times the inter-quartile range from the first and third quartiles.**

## 4.1 Evaluation against soil moisture

Data assimilation generally improved the agreement between modelled and field-measured top-layer soil moisture, increasing *r* by up to 0.3 compared to the open loop scenario (Fig. 4a). Based on Steiger tests (see Sect. 3.4), the improvements in model performance are statistically significant at 75–90 % of the sites, depending on the assimilated retrieval. Assimilating L-band soil moisture has the largest impact overall (average $\Delta r$ is 0.11), followed by the X-band (0.08) and C-band (0.04) retrievals, respectively. On a site-by-site basis, however, L-band soil moisture is not always the most informative. At 5 out of 24 sites (21 %), X-band retrievals are most informative, and at one site C-band retrievals are most informative (see Table S1). Modelled root-zone soil moisture also tends to improve after data assimilation, though model performance degrades slightly at about a quarter of the sites. The L-band and X-band retrievals have comparable results overall, though there is a slight advantage for L-band soil moisture on a site-by-site basis (most informative at 11 out of 24 sites, compared to 9 for X-band). Based on Steiger tests, about 90 % of the differences in *r* between these assimilation experiments are statistically significant.



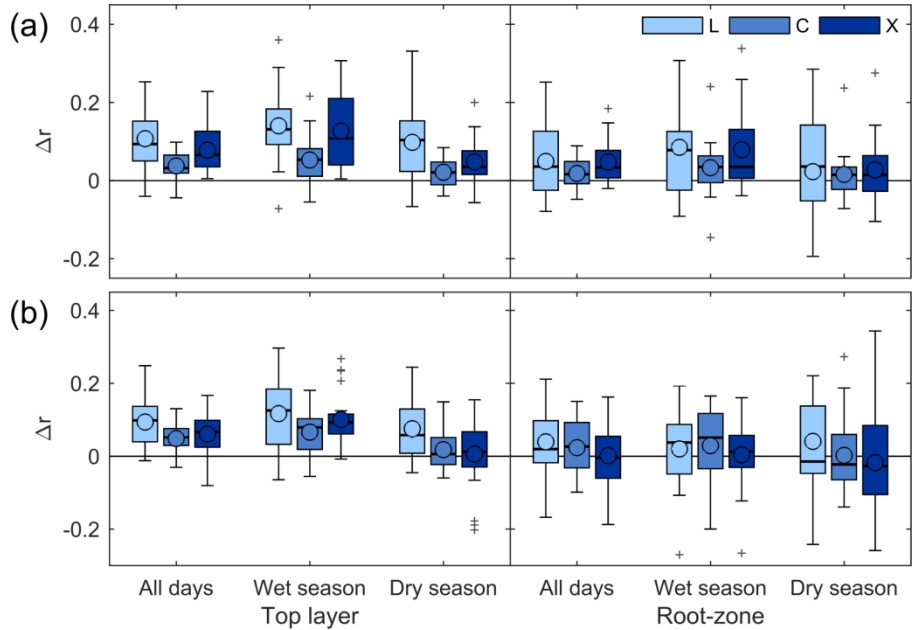

**Figure 4: The change in correlation (Δ*r*) between modelled and field-measured top-layer (N = 24) and root-zone (N = 19) soil moisture (July 2012 to April 2015) after soil moisture assimilation, where Δ*r* is calculated over the entire time period, the wet season and the dry season (a). In (b), the same is shown for anomaly time series. Box plots are defined as in Fig. 3.**

We examined the effect of season on the effectiveness of soil moisture assimilation by dividing the assimilation period into a wet season and dry season (see Sect. 3.5). For all three datasets, assimilation has a more positive effect on model performance of top-layer soil moisture in the wet season than in the dry season (Fig. 4a). The difference in model performance between the wet and dry seasons is largest for X-band assimilation and somewhat smaller for the other two datasets. In contrast, there is no difference between the dry and wet seasons in terms of the effect of data assimilation on root-zone soil moisture.

The effect of assimilation is also evaluated for anomaly time series, which reflects the model performance at sub-seasonal timescales. Similarly to the analysis based on actual time series, data assimilation tends to improve model performance against field-measured data (Fig. 4b). However, the magnitude of the changes is somewhat smaller and degradations in the agreement between model and field-measured root-zone soil moisture can be more substantial. For top-layer soil moisture, assimilating L-band retrievals leads to higher model performance than the other two datasets overall, and it is also the most informative at 13 out of 24 sites (54 %). In contrast to the evaluation of actual values, the C-band retrievals are most informative at more sites than X-band retrievals (at 8 compared to 3 out of 24 sites, respectively). On average, however, the changes in correlation are similar for both datasets (Fig. 4b). For root-zone soil moisture, the effectiveness of assimilation is similar between experiments. The added value of data assimilation for the top-layer is affected by the season, while this is not the case for the root zone.





Time series plots of modelled top-layer soil moisture at three sites (see Fig. 1 for their locations) illustrate the contrasting effect of assimilation in the wet and dry seasons. In wetter months, the model tends to overestimate soil moisture in the open loop scenario (Fig. 5). Data assimilation results in lower soil moisture content in this period, more closely matching the field-measured data. Periods where data assimilation leads to lower soil wetness also show lower evaporation rates. In drier months, however, the open loop simulation more closely matches the field-measured data, leaving little opportunity to improve the model. Sites where there is little to no rainfall in the dry season are particularly affected, as the assimilation tends to add noise to the otherwise smooth recession curves, as displayed by AU-Whr in Fig. 5. Furthermore, the time series show that the model error variance is reduced through the data assimilation.

**Figure 5:** **Time series of precipitation forcing and modelled evaporation and top-layer model soil moisture in the open loop (grey line and shading) and after assimilating L-band retrievals (blue line and shading) at sites Y10 (a), AU-Whr (b), AU-DaS (c). Time series of rescaled L-band retrievals (circles) and field–measured soil moisture (green line) are included for reference. These sites are classified as a grassland, savanna and evergreen broadleaf forest, respectively. Shaded areas represent the 10th-90th percentiles of the model ensemble.**





It is noted that there is a considerable difference in the number of observations in the soil moisture retrievals, with more observations in the C- and X-band datasets (662–830 observations in three years) than in the L-band dataset (225–500 observations). This difference in the number of observations could affect the impact of data assimilation, especially when evaluating the anomaly time series. However, correcting for the number and timing of observations by assimilating the

concurrent retrievals also used in joint assimilation has a limited effect on results. The relative impact of the soil moisture retrievals is unchanged, though model improvement after data assimilation is slightly lower. The lower impact of data assimilation is most likely because satellite observations, and thus model updates, are more infrequent. Fewer updates can result in a lower impact of data assimilation over a particular study period.

## 4.2 Added value of joint assimilation

Each combination of retrievals was assimilated simultaneously to assess whether there is complementary information in passive microwave bands of multiple frequencies. Joint assimilation of L- and C-band (or X-band) retrievals was superior to assimilating C-band (or X-band) retrievals individually, but performed similarly to assimilating L-band retrievals individually (Fig. 6a). On average, joint assimilation improved model performance compared to assimilating C-band (or X-band) individually by 0.07 (0.03), but on average, the difference between joint assimilation and assimilating L-band

retrievals individually was 0.00. Joint assimilation of C- and X-band retrievals improved performance compared to assimilating C-band retrievals individually, but overall slightly degraded model performance compared to assimilating X band individually. As a result, the sets including L-band retrievals as one of the two assimilated datasets outperform the joint assimilation of C-band and X-band retrievals. The difference between the combinations including L-band retrievals and the C- and X-band combination is statistically significant at nearly 90 % of the sites. When considering root-zone soil moisture,

the main difference was that the added value of joint assimilation of L- and C-band (or X-band) retrievals was smaller when compared to assimilating C-band (or X-band) retrievals individually.

Joint assimilation shows similar results when evaluating the anomaly time series. Again, joint assimilation of L-band soil moisture along with another dataset improved model performance compared to assimilating C- or X-band individually, but was not significantly different from assimilating L-band soil moisture individually (Fig. 6b). The joint assimilation of C- and

X-band retrievals further improved model agreement with field-measured data at nearly half of the sites, but the difference between joint assimilation and single assimilation of either dataset for this set is not statistically significant.





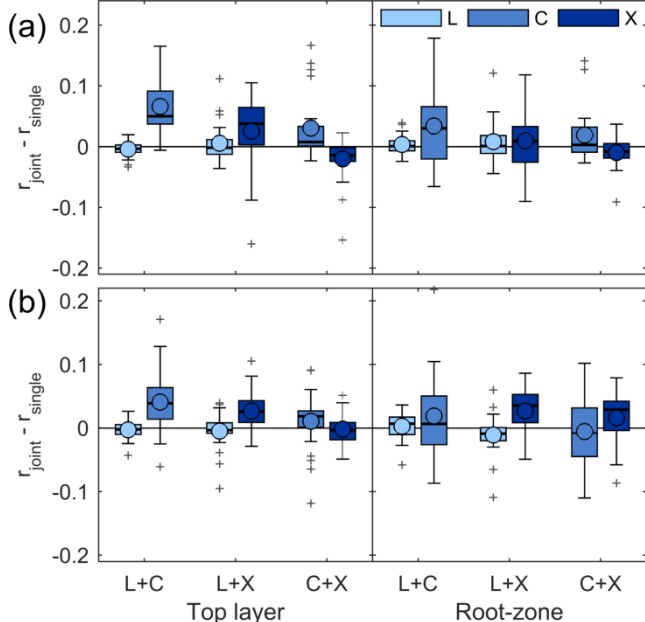

**Figure 6: The difference in model performance between the joint assimilation and single retrieval assimilation for surface (N = 24) and root-zone (N = 19) soil moisture (a), where model performance is based on the correlation (*r*) between modelled and field-measured soil moisture (July 2012 to April 2015). The legend indicates which single assimilation experiment is used as a reference to evaluate the joint assimilation experiments. In (b), the same is shown for anomaly time series. Box plots are defined as in Fig. 3.**

Overall, joint assimilation resulted in higher model performance than assimilating the less informative of the two retrievals, but did not substantially change model performance compared to the more informative retrieval. We use the effect of joint assimilation of L- and X-band retrievals on model top-layer soil moisture as an example. In the single retrieval assimilation experiments (Sect. 4.1), the L-band retrievals were more informative than X-band retrievals at 18 sites, X-band retrievals were more informative at 5 sites, and they were equally informative at one site (Table S1). On average, model performance of the joint assimilation experiment was the same as assimilating the more informative retrieval individually (average change in correlation was 0.00). However, model performance improved compared to assimilating the less informative retrieval (i.e. X-band retrievals at 18 sites and L-band retrievals at 5 sites) individually by an average of 0.05. This change in model performance is higher than when we use X-band assimilation as a reference (like in Fig. 6), which showed an average improvement of 0.03. Over all joint assimilation experiments, model performance improved compared to assimilating the less informative retrieval individually by 0.02–0.07 on average. Model performance did not change substantially compared to assimilating the more informative retrieval (−0.02–0.00).

## 4.3 Influence of precipitation data quality

We repeated the assimilation experiments using a higher-resolution and better quality precipitation dataset to evaluate the importance of the quality of the prior model estimates. At all but three sites, forcing AWRA-L with AGCD precipitation leads to better open loop model performance than when TRMM precipitation is used (Fig. 7a). The *r* between model and



field-measured soil moisture differs by up to 0.3. The three sites where AGCD precipitation leads to lower model performance are located in northern Australia, where the density of precipitation gauges used to create the AGCD precipitation dataset is relatively low. The largest improvements in model performance when using AGCD instead of TRMM precipitation are found in south-eastern Australia, where the gauge density is relatively high.

The impact of data assimilation was generally higher when the model was forced with lower-quality TRMM precipitation data. In other words, the change in model performance after data assimilation ($\Delta r$) was larger for the TRMM experiments than for the AGCD experiments (Fig. 7b). The difference was relatively large for X- and L-band retrievals, and relatively small when assimilating C-band retrievals. On average, data assimilation improved correlations with field-measured by 0.01–0.05 more when the model was forced with TRMM precipitation than when forced with AGCD precipitation,

depending on which retrievals were assimilated (0.01–0.04 for root-zone data). Despite the fact that data assimilation is less informative when AGCD precipitation is used, it generally still has a positive impact on model performance. Correlations between model and field-measured soil moisture increased by an average of 0.02–0.07 for top-layer soil moisture and 0.01–0.04 for top-layer soil moisture, depending on the assimilated retrieval(s).

   Precipitation quality had a larger effect on model open loop performance based on anomaly time series (Fig. 7c) than based

on actual time series. On average, the difference in the impact of data assimilation between precipitation datasets is similar for both precipitation datasets (Fig. 7d). However, the variability in the difference in the impact of data assimilation was higher for anomaly time series.

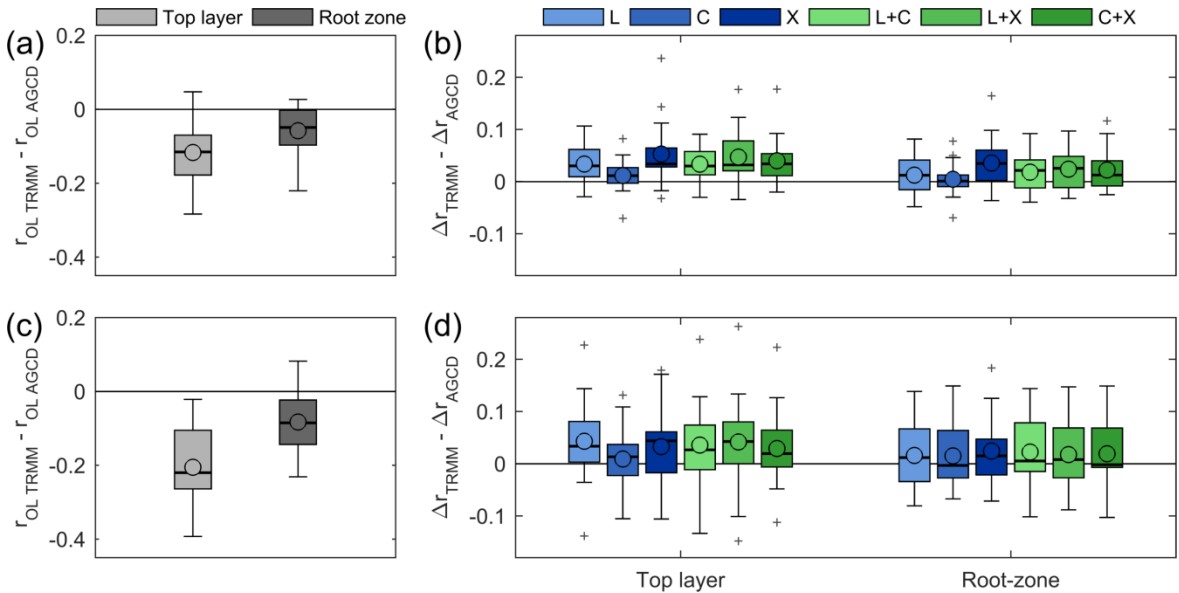

**Figure 7: The difference in open loop model performance based on *r* (a) and the difference between the impact of data assimilation based on *Δr* (b) between experiments using TRMM and AGCD precipitation data. Panels (c) and (d) show the same for anomaly time series. Box plots are defined as in Fig. 3.**



If we use the difference in open loop model performance as a proxy for precipitation quality, we can quantify the relationship between the difference in precipitation quality of the TRMM and AGCD datasets (i.e. data in Fig. 7a and c) and the corresponding difference in the impact of data assimilation (i.e. data in Fig. 7b and d). Pearson's $r$ between precipitation data quality and impact of data assimilation experiments vary between −0.83 and −0.57 for top-layer soil moisture, except when assimilating C-band retrievals individually ($r = −0.22$). The relationship for root-zone soil moisture is slightly weaker, with $r$ between −0.79 and −0.32 (except $r = −0.16$ for C band). This means that where using AGCD and TRMM precipitation resulted in similar model performance, the impact of data assimilation on model performance was also similar. Where using AGCD precipitation resulted in higher model performance than TRMM, on the other hand, data assimilation had a smaller impact than when TRMM precipitation was used.

## 5 Discussion

We approached this study of the impact of assimilating satellite soil moisture retrievals based on different microwave frequencies from an empirical perspective. As discussed in Sect. 3.5, differences in the results of the assimilation experiments are affected not only by frequency, but also by differences in the characteristics of the sensors and platforms, in the parameterization of the common retrieval model, and in the error characteristics of the retrievals. A more theoretical approach is needed to untangle these confounding factors.

Another important choice in the experimental setup was the rescaling technique. We chose a relatively simple linear rescaling technique to transform satellite observations to model space. Previous studies have suggested that more sophisticated bias correction techniques are more suitable (i.e. Yilmaz and Crow 2013), but these techniques usually assume that there is no consistent bias between the model and observations. Here, we found that the model was clearly biased in the wet season, especially for AU-DaS (Fig. 5). In those conditions, observations have considerable potential to improve model results, and that potential would be diminished by forcing the observations to match the incorrect model dynamics.

Overall, L-band soil moisture retrievals showed the best agreement with field-measured data over the study sites (Fig. 3), which is consistent with previous studies (Al-Yaari et al., 2014; Holgate et al., 2016). Interestingly, errors did not increase with increasing frequency, and the X-band retrievals performed better than C-band retrievals. This may be typical of the sites studied here or of the AMSR-2 sensor, as previous studies have found that LPRM C-band retrievals from AMSR-2's predecessor, AMSR-E, slightly outperform X-band retrievals (Gruhier et al., 2009; Parinussa et al., 2011). As expected, datasets with smaller errors were generally more informative in assimilation, especially for top-layer soil moisture. This can be attributed in part to the differences in the magnitude of the errors, as this affects the weight given to the observations in the assimilation procedure. On average, triple collocation errors for C-band retrievals were 0.24 (AWRA-L wetness units), compared to 0.18 for the other retrievals. Further research is needed to evaluate whether these differences in errors are due to the trade-off between spatial resolution and sensitivity to vegetation/atmosphere or whether they are the result of other factors. For the root zone, differences between the assimilation experiments are much less pronounced (Fig. 4). The similar





information content in L- and X-band retrievals, especially, implies that data assimilation systems can substitute one retrieval for the other without substantially affecting model performance. This is especially important for modeling systems that cover a relatively long time period that need to transition between microwave sensors and missions.

Joint assimilation of two passive microwave soil moisture retrievals usually resulted in model performance that was similar to assimilating the more informative of the two retrievals individually, and improved model performance compared to assimilating the less informative retrieval alone (by 0.02–0.07 on average, Fig. 6 and Sect. 4.3). Therefore, joint assimilation appears especially useful to take advantage of the superior information in whichever retrieval is most informative in a particular location, without substantially degrading model performance by the retrievals that are less informative. This means that joint assimilation is of added value when no single retrieval is most informative in the study area, as was the case in this study. However, joint assimilation may not be of added value in studies where one soil moisture retrieval is most informative. Based on the results of this study, combining L-band retrievals (SMOS) with either the C- or X-band retrievals (AMSR-2) is most informative. However, the joint assimilation of C- and X-band retrievals performs surprisingly well considering the fact that errors between these retrievals, derived from the same AMSR-2 sensor, might not be expected to be fully independent. Theoretically, this would compound errors in data assimilation, and degrade performance, but this was not usually observed. Since we corrected for the number and timing of observations, this added value of joint assimilation can be attributed to other factors, which may include error characteristics, and characteristics of the sensors and microwave frequencies such as observation depth, spatial resolution, and viewing angle.

Finally, the quality of meteorological data, and precipitation, in particular, is an important driver of the performance of hydrological models. Top-layer soil moisture was more sensitive to using different quality precipitation datasets than root-zone soil moisture (Fig. 7), which may indicate that errors between precipitation datasets are attenuated in the root zone. Data assimilation is more informative when the quality of precipitation datasets is relatively low, suggesting that assimilation is able to (partly) correct errors between low- and high-quality precipitation datasets. However, data assimilation is still worthwhile when high-quality precipitation datasets are used (Sect. 4.3). This is despite the fact that the soil moisture retrievals have relatively coarse resolutions compared to the resolution of the model and AGCD forcing datasets.

This study only examined the temporal aspect of differences and complementary information between the different datasets. The model was run for each site independently and did not consider spatial covariance. It is conceivable that including this spatial dimension could change the results, for example improving the added value of joint assimilation. The complementary spatial information in passive microwave retrievals could stem from the differences in the native footprint sizes of the different microwave frequencies, or from using sharpened soil moisture retrievals (i.e. Merlin et al. 2013; Piles et al. 2011; Kim and Hogue 2012; Gevaert et al. 2015). The spatial information could improve the spatial patterns of soil moisture in models, which could propagate into improved simulations of runoff and streamflow.



## 6 Summary and conclusions

Passive microwave soil moisture retrievals based on different frequencies but derived by a common retrieval algorithm were assimilated into the AWRA-L model. Model results were evaluated against field-measured soil moisture at 24 sites spread over the Australian continent to compare the ability. This evaluation compares the ability of the datasets to improve model
soil moisture through data assimilation and assesses whether there is added value in joint assimilation. The study sites cover a range of climate and land cover types, but the evaluation is complicated by the differences in the representative area of the model and field-measured data.

Data assimilation generally has a positive impact on the performance of model top-layer and root-zone soil moisture, increasing $r$ with field-measured data by up to 0.3. Assimilation improves model performance more in the wet season, when
the model skill is relatively low, than in the dry season, when the model skill is relatively high. The impact of data assimilation is also higher when the model is forced with global precipitation data when it is forced with higher-quality, higher-resolution precipitation data.

Overall, assimilating L- and X-band retrievals had a more positive impact on model performance than assimilating C-band retrievals. L-band retrievals are slightly more informative than X-band retrievals when evaluating top-layer soil moisture, but
these differences are not statistically significant and the advantage disappears when root-zone soil moisture is considered. The large overall consistency between assimilating L- or X-band retrievals, and even C-band retrievals in the root zone, implies that assimilation studies can temporarily or permanently switch between these retrievals with little to no effect on model performance. This is particularly advantageous for studies and applications which assimilate soil moisture over long periods of time and are thus obliged to use retrievals from different sensors and platforms.

When two passive microwave retrievals are assimilated simultaneously, model performance is similar to or better than assimilating either of the bands individually, especially when combining an AMSR-2 retrieval (C- or X-band) with SMOS retrievals (L-band). This means that joint assimilation can be of added value when different soil moisture retrievals are more informative in different locations. It is likely that including spatial aspects would increase the added value of joint assimilation due to the trade-off in passive microwave soil moisture retrievals between footprint size on the one hand and
observation depth and sensitivity to vegetation and the atmosphere on the other. Additional studies focusing on spatial patterns of soil moisture are needed to quantify the complementary spatial information in passive microwave retrievals, whether at their native resolution or masking use of sharpened soil moisture datasets.

**Acknowledgments**

This research received funding from the European Union Seventh Framework Programme (FP7/2007-2013) under grant
agreement no. 603608, Global Earth Observation for integrated water resource assessment: eartH2Observe. A.I.J.M. van Dijk was supported under Australian Research Council's Discovery Projects funding scheme (project DP140103679). The authors would like to thank Robin van der Schalie for providing the SMOS data. This work also used eddy covariance data





collected by the TERN-OzFlux facility. OzFlux would like to acknowledge the financial support of the Australian Federal Government via the National Collaborative Research Infrastructure Scheme and the Education Investment Fund.

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
