# Peer review of "Joint assimilation of soil moisture retrieved from multiple passive microwave frequencies increases robustness of soil moisture state estimation"

_Hydrology and Earth System Sciences, 2017_

## Referee Comment (RC1) · Anonymous Referee #1 · 14 Apr 2018

**OVERVIEW**

The manuscript investigates the potential of assimilating passive microwave soil moisture observations in land surface modelling for improving surface and root-zone soil moisture simulation. Specifically, the assimilation of L-, C-, and X-band observations obtained by SMOS and AMSR2 satellites into the AWRA-L model in Australia is carried out. In situ observations at surface and root-zone are used for assessing the impact of

satellite soil moisture assimilation.

**GENERAL COMMENTS**

The manuscript is well written and clear. The topic is surely of interest for the readership of HESS as the availability of multiple soil moisture products from satellite sensors poses the question on how to fully exploit their combination/integration for improving hydrological applications. The study investigates this aspect by using passive-based microwave retrievals at L-, C-, and X-band. Therefore, I believe the paper deserves to be published. The paper reads well and I have only a couple of suggestions that can be implemented to improve further, in my opinion, the significance of the paper.

1) Two precipitation products are considered to assess the influence of their quality on soil moisture assimilation. It was recently done also in Massari et al. (2018, doi: 10.3390/rs10020292), and I believe it is a very interesting approach. However, a gauge-corrected product (research version of TRMM) and a gauge-based product are considered. I would suggest to consider also the real-time version of TRMM, better of TMPA, that is only satellite-based and would provide information of the impact of soil moisture assimilation in areas of the world in which raingauge are not present. I believe that it would not be too much work to be done, and the corresponding results would be of strong interest (at least for me).

2) The analysis has split the assimilation in wet and dry periods, as in wet conditions AWRA-L model is not performing well. However, in many previous studies it was obtained that the higher impact of soil moisture assimilation is obtained in the transition periods between dry and wet conditions (and viceversa between wet and dry conditions). I would suggest including these periods in the analysis (again, it should be easy to be added).

**SPECIFIC COMMENTS**

Page 10, figure 3: The performance of the Open Loop simulation for root-zone soil moisture simulation should be added

Page 11, line 9: "no difference". I suggest changing with "a small difference" or equivalent, as some differences are present also for root-zone soil moisture.

Page 16, Discussion section: I suggest adding a paragraph of comparison with studies that have considered the joint assimilation of active and passive soil moisture products, to highlight the similarities and the differences.

**RECCOMMENDATION**

On this basis, I found the topic of the paper relevant and interesting. Therefore, I suggest a minor revision before the publication in Hydrology and Earth System Sciences.

**REFERENCES**

Massari, C., Camici, S., Ciabatta, L., Brocca, L. (2018). Exploiting satellite-based surface soil moisture for flood forecasting in the Mediterranean area: state update versus rainfall correction. Remote Sensing, 10(2), 292, doi: 10.3390/rs10020292. http://dx.doi.org/10.3390/rs10020292.

---

## Referee Comment (RC2) · J. M. Bergeron (Referee) · 14 May 2018

The manuscript provides useful insight into the added value of various combinations of soil moisture retrievals from different passive microwave frequencies by assimilating them individually and jointly into a hydrological model and comparing the results with in situ measurements. The manuscript is very well written, using clear language and properly justifying the use of the various methods used throughout the study. The method is sound and the discussion is thorough. Good work!

[Figure]

I have a few questions and minor suggestions to further improve the manuscript.

My first question concerns the inflation factor. This inflation factor (let's call it gamma) was applied increase the ensemble spread to avoid having a disproportionate ratio of model and observation errors, which would lead to observations having no impact on the model analysis. From my understanding of the text (p.7, l.15-16), the actual value of this inflation factor is not specified, but is chosen in such a manner as to avoid the model error (for both top layer and root zone?) from ever falling below 2%. Is this correct? In this case, what is the value of gamma?

My second question relates to variables boundaries. The inflation factor works well for unbounded variables, but problems may arise when an ensemble member approaches a boundary. Ensuring physical realism (e.g. by adjusting negative values to zero) may introduce a bias. How are boundaries handled for modelled variables?

On a similar note, how do you perturb the observations? The errors attributed to the C-band retrievals is said to be 0.24 in AWRA-L wetness units and 0.18 for the other retrievals (p.16, l.29-30). I am assuming these are standard deviations? Either way, in the original EnKF procedure, the observations are perturbed to reach these errors. What type of distribution is used and how are observation boundaries handled? Using a Gaussian distribution on a bounded variable leads to values falling outside the boundaries. Also, observation values of 0% or 100% cannot be perturbed without introducing a bias. If this is the case, I believe a discussion on the matter would be beneficial.

As for suggestions, the first concerns the title, which I believe is misleading. The manuscript showed there was little added value to the joint assimilation schemes compared with assimilating individual bands. While the possibility of added value was mentioned where individual soil moisture retrievals are more informative in different locations

I would also recommend mentioning the ensemble Kalman filter somewhere in the abstract as it is an important part of the method and would facilitate the search for

relevant information for readers.

---

## Author Comment (AC1) · 26 Jun 2018

Comment #1

Two precipitation products are considered to assess the influence of their quality on soil moisture assimilation. It was recently done also in Massari et al. (2018, doi: 10.3390/rs10020292), and I believe it is a very interesting approach. However, a gauge-corrected product (research version of TRMM) and a gauge-based product are considered. I would suggest to consider also the real-time version of TRMM, better of

TMPA, that is only satellite-based and would provide information of the impact of soil moisture assimilation in areas of the world in which raingauge are not present. I believe that it would not be too much work to be done, and the corresponding results would be of strong interest (at least for me).

Response #1

The results presented in our study are in fact based on the real-time version of TRMM, though we now see that the text wrongly refers to the research version. As you explain, by using the real-time version we can assess what the benefits of (joint) assimilation of the soil moisture products would be where good-quality precipitation data is not available.

We have also assessed how model quality differs when using the research or real-time version of TRMM (Fig. 1). When using the real-time version, the agreement between top layer soil moisture and observational data is a bit lower than when the research version is used (by 0.04 on average), as expected. The quality of root-zone soil moisture is similar for both models. However, even for the top layer the difference in the quality of the TRMM precipitation products is relatively small when model results using AWAP gauge-based precipitation are used as a reference. As a result, the impact of soil moisture assimilation based on $\Delta r$ is very similar for both TRMM precipitation products. On average, $\Delta r$ is 0.01 to 0.03 larger when the real-time version of TRMM is used than when the research version is used, depending on the soil moisture product assimilated.

In the revised version of the manuscript, we will make the choice for real-time TRMM clearer.

Comment #2

The analysis has split the assimilation in wet and dry periods, as in wet conditions AWRA-L model is not performing well. However, in many previous studies it was obtained that the higher impact of soil moisture assimilation is obtained in the transition periods between dry and wet conditions (and viceversa between wet and dry conditions). I would suggest including these periods in the analysis (again, it should be easy to be added).

Response #2

Thank you for your suggestion. We have analyzed the impact of assimilation for transitional conditions. However, we have found that assimilation is not necessarily more informative in these months than in the (shortened) wet and dry periods (Fig. 2 below). In the top layer, the impact of assimilation in the transitional period compared to the wet and dry seasons depends on the retrieval. For L-band, the impact of assimilation is highest in the transitional period (on average). For C-band, however, it is lowest, and for X-band the impact is higher than the dry season, but lower than the wet season. For root-zone soil moisture, the impact of assimilation in the transition and wet seasons is similar (except for C band) and higher than in the dry season. Based on anomaly time series, assimilation is only informative in the top layer during transitional periods, and not for the root zone.

In the revised version, we will evaluate the impact of soil moisture assimilation for the transitional periods between the wet and dry seasons as well. Therefore, Fig. 4 will be replaced by Figure R2. The text referring to the seasonality of the impact of assimilation will be updated to reflect the new results.

SPECIFIC COMMENTS

R3) Page 10, figure 3: The performance of the Open Loop simulation for root-zone soil moisture simulation should be added

A3) Agreed, we will add the performance of the open loop for the root zone to Figure 3.

R4) Page 11, line 9: "no difference". I suggest changing with "a small difference" or equivalent, as some differences are present also for root-zone soil moisture.

[Figure]

A4) Agreed, this will be changed in the revised manuscript.

R5) Page 16, Discussion section: I suggest adding a paragraph of comparison with studies that have considered the joint assimilation of active and passive soil moisture products, to highlight the similarities and the differences.

A5) In the introduction of our study we reference two studies that jointly assimilated an active (both ASCAT) and a passive (both AMSR-E) soil moisture retrieval into land surface models, Draper et al. (2012) and Renzullo et al. (2014). Both of those studies are also (partly) based on Australian sites, resulting in some overlap with our study sites.

Draper et al. (2012) assimilated an active and a passive soil moisture retrieval into a land surface model and evaluated model performance over a number of sites in the US and the Murrumbidgee catchment in Australia. Their conclusion was similar to our own, namely that joint assimilation led to similar or better model performance than assimilating either product individually. In contrast, Renzullo et al., (2014) stated that joint assimilation resulted in a compromise between the two retrievals at a number of study sites spread around Australia. However, where reported, the correlations of the joint assimilation experiments were at most 0.02 lower than when assimilating the more informative soil moisture product individually (Renzullo et al., 2014), suggesting that the performance is largely similar. In the revised manuscript we will add a comparison of the results of the joint assimilation experiments to those of Draper et al. (2012) and Renzullo et al. (2014).

RECCOMMENDATION

R) On this basis, I found the topic of the paper relevant and interesting. Therefore, I suggest a minor revision before the publication in Hydrology and Earth System Sciences.

A) Thank you.

REFERENCES Massari, C., Camici, S., Ciabatta, L., Brocca, L. (2018). Exploiting satellite-based surface soil moisture for flood forecasting in the Mediterranean area: state update versus rainfall correction. Remote Sensing, 10(2), 292, doi: 10.3390/rs10020292. http://dx.doi.org/10.3390/rs10020292.

Draper, C. S., Reichle, R. H., De Lannoy, G. J. M. and Liu, Q.: Assimilation of passive and active microwave soil moisture retrievals, Geophys. Res. Lett., 39(4), L04401, doi:10.1029/2011GL050655, 2012.

Renzullo, L. J., van Dijk, A. I. J. M., Perraud, J.-M., Collins, D., Henderson, B., Jin, H., Smith, A. B. and McJannet, D. L.: Continental satellite soil moisture data assimilation improves root-zone moisture analysis for water resources assessment, J. Hydrol., 519, 2747–2762, doi:10.1016/j.jhydrol.2014.08.008, 2014.
* * *
[Figure]

**Fig. 1.** Correlations between model and field-measured soil moisture for the top layer and root zone when using different precipitation products based on actual (left) and anomaly (right) time series.
(a)

(b)

**Fig. 2.** The change in correlation (△r) between modelled and field-measured top-layer and root-zone soil moisture after soil moisture assimilation based on actual (a) and anomaly (b) time series.

---

## Author Response (AR1)

We would like to thank the referees for their valuable and constructive comments. The feedback has helped us improve the manuscript. Below, we respond to the comments in a point-by-point fashion and indicate what changes have been made to the manuscript.

**Anonymous referee 1**

**OVERVIEW**

The manuscript investigates the potential of assimilating passive microwave soil moisture observations in land surface modelling for improving surface and root-zone soil moisture simulation. Specifically, the assimilation of L-, C-, and X-band observations obtained by SMOS and AMSR2 satellites into the AWRA-L model in Australia is carried out. In situ observations at surface and root-zone are used for assessing the impact of satellite soil moisture assimilation.

**GENERAL COMMENTS**

The manuscript is well written and clear. The topic is surely of interest for the readership of HESS as the availability of multiple soil moisture products from satellite sensors poses the question on how to fully exploit their combination/integration for improving hydrological applications. The study investigates this aspect by using passive-based microwave retrievals at L-, C-, and X-band. Therefore, I believe the paper deserves to be published. The paper reads well and I have only a couple of suggestions that can be implemented to improve further, in my opinion, the significance of the paper.

**Comment #1**

Two precipitation products are considered to assess the influence of their quality on soil moisture assimilation. It was recently done also in Massari et al. (2018, doi: 10.3390/rs10020292), and I believe it is a very interesting approach. However, a gauge-corrected product (research version of TRMM) and a gauge-based product are considered. I would suggest to consider also the real-time version of TRMM, better of TMPA, that is only satellite-based and would provide information of the impact of soil moisture assimilation in areas of the world in which raingauge are not present. I believe that it would not be too much work to be done, and the corresponding results would be of strong interest (at least for me).

**Response #1**

The results presented in our study are in fact based on the real-time version of TRMM, though we now see that the original manuscript wrongly refers to the research version. As you explain, by using the real-time version we can assess what the benefits of (joint) assimilation of the soil moisture products would be where good-quality precipitation data is not available.

We have also assessed how model quality differs when using the research or real-time version of TRMM (Fig. R1). When using the real-time version, the agreement between top layer soil moisture and observational data is a bit lower than when the research version is used (by 0.04 on average), as expected. The quality of root-zone soil moisture is similar for both models. However, even for the top layer the difference in the quality of the TRMM

precipitation products is relatively small when model results using AWAP gauge-based precipitation are used as a reference. As a result, the impact of soil moisture assimilation based on  $\Delta r$  is very similar for both TRMM precipitation products. On average,  $\Delta r$  is 0.01 to 0.03 larger when the real-time version of TRMM is used than when the research version is used, depending on the soil moisture product assimilated.

In the revised version of the manuscript, we clarify that the real-time version is used (P5 L8) and clarify our motivation to use it as follows (P9 L7-9):

"By using the real-time version of TRMM, which is not gauge-corrected, we can assess the added value of soil moisture assimilation in regions where there are fewer rain gauges and precipitation data quality is therefore relatively poor."

**Fig. R1. Correlations between model and field-measured soil moisture for the top layer and root zone when using different precipitation products. Correlations are based on actual (left) and anomaly (right) time series. TRMMRT is the real-time version of TRMM, TRMMRV the research version.**

**Comment #2**

The analysis has split the assimilation in wet and dry periods, as in wet conditions AWRA-L model is not performing well. However, in many previous studies it was obtained that the higher impact of soil moisture assimilation is obtained in the transition periods between dry and wet conditions (and viceversa between wet and dry conditions). I would suggest including these periods in the analysis (again, it should be easy to be added).

**Response #2**

Thank you for your suggestion. We have analyzed the impact of assimilation for transitional conditions. However, we have found that assimilation is not necessarily more informative in these months than in the (shortened) wet and dry periods (Fig. R2). In the top layer, the impact of assimilation in the transitional period compared to the wet and dry seasons depends on the retrieval. For L-band, the impact of assimilation is highest in the transitional period (on average). For C-band, however, it is lowest, and for X-band the impact is higher than the dry season, but lower than the wet season. For root-zone soil moisture, the impact of assimilation in the transition and wet seasons is similar (except for C band) and higher than in the dry season. Based on anomaly time series, assimilation is only informative in the top layer during transitional periods, and not for the root zone.

In the revised version, we evaluate the impact of soil moisture assimilation for the transitional periods between the wet and dry seasons as well. Specifically, Fig. 4 of the

manuscript has been replaced by Figure R2 and the text referring to the seasonality of the impact of assimilation has been updated to reflect the new results as discussed in the previous paragraph (P11 L8-16 and P12 L2-4).

Fig. R2. The change in correlation ( $\Delta r$ ) between modelled and field-measured top-layer and root-zone soil moisture after soil moisture assimilation, where  $\Delta r$  is calculated over the entire time period, as well as divided into the wet and dry seasons, as well as the transitional periods (a). In (b), the same is shown for anomaly time series.

**SPECIFIC COMMENTS**

Page 10, figure 3: The performance of the Open Loop simulation for root-zone soil moisture simulation should be added

Agreed, we have added the performance of the open loop for the root zone to Figure 3 in the revised manuscript.

Page 11, line 9: "no difference". I suggest changing with "a small difference" or equivalent, as some differences are present also for root-zone soil moisture.

Agreed, this has been changed in the revised manuscript.

Page 16, Discussion section: I suggest adding a paragraph of comparison with studies that have considered the joint assimilation of active and passive soil moisture products, to highlight the similarities and the differences.

In the introduction of our study we reference two studies that jointly assimilated an active (both ASCAT) and a passive (both AMSR-E) soil moisture retrieval into land surface models, Draper et al. (2012) and Renzullo et al. (2014). Both of those studies are also (partly) based on Australian sites, resulting in some overlap with our study sites. The following paragraph discussing these papers has been added to the discussion section in the revised manuscript (P18 L18-29):

"Though we are not aware of other studies jointly assimilating passive soil moisture retrievals, our results are in line with studies jointly assimilating active and passive soil moisture retrievals. Two studies, in particular, also use Australian study sites to evaluate the impact of (joint) soil moisture assimilation into land surface models. Draper et al. (2012) evaluated soil moisture assimilation at sites in the United States and south-eastern Australia, while Renzullo et al. (2014) focused on sites spread around Australia as in this study. The active and passive soil moisture retrievals were based on C-bane microwave data in both studies and therefore focus on complementary information in retrieval method, while in this study we focus on added value between microwave frequencies. Nevertheless, the conclusion of Draper et al. (2012) is very similar to that of this study, namely that joint assimilation leads to similar or better model performance than assimilating either retrieval individually. In contrast, Renzullo et al. (2014) stated that joint assimilation resulted in a compromise between the two retrievals. However, where reported, the correlations of the joint assimilation experiments were at most 0.02 lower than when assimilating the more informative soil moisture product individually in that study (Renzullo et al., 2014), suggesting that model performance is in fact similar."

**RECCOMMENDATION**

On this basis, I found the topic of the paper relevant and interesting. Therefore, I suggest a minor revision before the publication in Hydrology and Earth System Sciences.

Thank you.

The manuscript provides useful insight into the added value of various combinations of soil moisture retrievals from different passive microwave frequencies by assimilating them individually and jointly into a hydrological model and comparing the results with in situ measurements. The manuscript is very well written, using clear language and properly justifying the use of the various methods used throughout the study. The method is sound and the discussion is thorough. Good work!

I have a few questions and minor suggestions to further improve the manuscript.

**Comment #1**

My first question concerns the inflation factor. This inflation factor (let's call it gamma) was applied increase the ensemble spread to avoid having a disproportionate ratio of model and observation errors, which would lead to observations having no impact on the model analysis. From my understanding of the text (p.7, l.15-16), the actual value of this inflation factor is not specified, but is chosen in such a manner as to avoid the model error (for both top layer and root zone?) from ever falling below 2%. Is this correct? In this case, what is the value of gamma?

**Response #1**

The inflation factor ensures that the model error does not fall below 2 %. The model error is based on the model error variance  $H_tP_tH_t^T$ , which is based on the top layer of the model since this is the layer that is most representative of the shallow depth of the soil moisture observations. If we would have chosen a fixed inflation factor the 2 % error would only be ensured with a (very) large value. Therefore, we used a variable inflation factor. This inflation factor was calculated at each timestep where the model error was calculated as being smaller than 2 % by the equation:

$$\gamma = \sqrt{0.02^2/(H_t P_t H_t^T)}$$

In the revised manuscript, we have added the following text to address this point (P7 L16-18):

"The variable inflation factor is applied only at time steps when the model error of the top layer falls below this value, and its magnitude is determined by the ratio between the desired and calculated model variance  $(H_t P_t H_t^T)$  at that time step."

**Comment #2**

My second question relates to variables boundaries. The inflation factor works well for unbounded variables, but problems may arise when an ensemble member approaches a boundary. Ensuring physical realism (e.g. by adjusting negative values to zero) may introduce a bias. How are boundaries handled for modelled variables?

**Response #2**

It is true that using an inflation factor on a bounded variable will introduce bias when the values are adjusted to its boundaries. However, to avoid using unphysical values, we adjusted

negative values to zero and values above the storage capacity of a soil layer to its maximum value immediately after applying the inflation factor. This adjustment, being one-sided, will introduce bias in the modeled time series. However, as discussed in response to the previous comment, the inflation factor was only applied when the model error fell below 2 % and its magnitude was set to ensure a model error of 2 %. Therefore, even in the worst case scenario, when the moisture store is predicted to be at capacity or empty, the magnitude of the bias is limited.

A summary of this response has been included in the revised manuscript, explaining the consequences of the inflation factor on model time series when the values approach the physical boundaries of the soil moisture layers (P7 L18-23).

**Comment #3**

On a similar note, how do you perturb the observations? The errors attributed to the C-band retrievals is said to be 0.24 in AWRA-L wetness units and 0.18 for the other retrievals (p.16, l.29-30). I am assuming these are standard deviations? Either way, in the original EnKF procedure, the observations are perturbed to reach these errors. What type of distribution is used and how are observation boundaries handled? Using a Gaussian distribution on a bounded variable leads to values falling outside the boundaries. Also, observation values of 0% or 100% cannot be perturbed without introducing a bias. If this is the case, I believe a discussion on the matter would be beneficial.

**Response #3**

The observations were perturbed using a normal distribution, with the error values representing the standard deviation. As you mentioned, this can indeed lead to values falling outside the range of possible values in bounded variables such as soil moisture. Therefore, the values were adjusted to fall within the physical range: negative values were set to zero, values larger than one to one. This adjustment can indeed lead to bias.

In the revised manuscript, we have added the following explanation (P 8 L1-5):

"The observations were perturbed according to a Gaussian distribution with the triple collocation error estimates as standard deviations. Similarly to the variable inflation factor, perturbing soil moisture observations near the boundaries of the variables may result in values that are not physically real. The values falling outside the boundaries are therefore adjusted to the nearest limit. However, this process may introduce bias, especially where soil moisture observations are near its upper and lower boundaries."

**Comment #4**

As for suggestions, the first concerns the title, which I believe is misleading. The manuscript showed there was little added value to the joint assimilation schemes compared with assimilating individual bands. While the possibility of added value was mentioned where individual soil moisture retrievals are more informative in different locations.

Response #4

Thank you for your suggestion. We have changed the title to "Joint assimilation of soil moisture retrieved from multiple passive microwave frequencies increases robustness of soil moisture state estimation".

**Comment #5**

I would also recommend mentioning the ensemble Kalman filter somewhere in the abstract as it is an important part of the method and would facilitate the search for relevant information for readers.

**Response #5**

Agreed, we have added the fact we used an ensemble Kalman filter to the abstract (P1, L17).

**Joint assimilation of soil moisture retrieved from multiple passive microwave frequencies increases robustness and quality of soil moisture state estimation**

Anouk I. Gevaert1, Luigi J. Renzullo2,3, Albert I. J. M. van Dijk23, Hans J. van der Woerd34, Albrecht H. Weerts45,65, Richard A. M. de Jeu67

[revised manuscript text omitted]